# Candidate Biomarkers for Specific Intraoperative Near-Infrared Imaging of Soft Tissue Sarcomas: A Systematic Review

**DOI:** 10.3390/cancers13030557

**Published:** 2021-02-01

**Authors:** Zeger Rijs, A. Naweed Shifai, Sarah E. Bosma, Peter J. K. Kuppen, Alexander L. Vahrmeijer, Stijn Keereweer, Judith V. M. G. Bovée, Michiel A. J. van de Sande, Cornelis F. M. Sier, Pieter B. A. A. van Driel

**Affiliations:** 1Department of Orthopedic Surgery, Leiden University Medical Center, Albinusdreef 2, 2333 ZA Leiden, The Netherlands; A.N.Shifai@lumc.nl (A.N.S.); S.E.Bosma@lumc.nl (S.E.B.); M.A.J.van_de_Sande@lumc.nl (M.A.J.v.d.S.); 2Department of Surgery, Leiden University Medical Center, Albinusdreef 2, 2333 ZA Leiden, The Netherlands; P.J.K.Kuppen@lumc.nl (P.J.K.K.); A.L.Vahrmeijer@lumc.nl (A.L.V.); C.F.M.Sier@lumc.nl (C.F.M.S.); 3Department of Otorhinolaryngology Head and Neck Surgery, Erasmus Medical Center Cancer Institute, Wytemaweg 80, 3015 CN Rotterdam, The Netherlands; S.Keereweer@erasmusmc.nl; 4Department of Pathology, Leiden University Medical Center, Albinusdreef 2, 2333 ZA Leiden, The Netherlands; J.V.M.G.Bovee@lumc.nl; 5Percuros BV, Zernikedreef 8, 2333 CL Leiden, The Netherlands; 6Department of Orthopedic Surgery, Isala Hospital, Dokter van Heesweg 2, 8025 AB Zwolle, The Netherlands; p.b.a.a.van.driel@isala.nl

**Keywords:** TEM1, VEGFR-1, EGFR, VEGFR-2, IGF-1R, PDGFRα, CD40, image guided surgery, near-infra red fluorescence, soft tissue sarcomas

## Abstract

**Simple Summary:**

Near-infrared imaging of tumors during surgery facilitates the oncologic surgeon to distinguish malignant from healthy tissue. The technique is based on fluorescent tracers binding to tumor biomarkers on malignant cells. Currently, there are no clinically available fluorescent tracers that specifically target soft tissue sarcomas. This review searched the literature to find candidate biomarkers for soft tissue sarcomas, based on clinically used therapeutic antibodies. The search revealed 7 biomarkers: TEM1, VEGFR-1, EGFR, VEGFR-2, IGF-1R, PDGFRα, and CD40. These biomarkers are abundantly present on soft tissue sarcoma tumor cells and are already being targeted with humanized monoclonal antibodies. The conjugation of these antibodies with a fluorescent dye will yield in specific tracers for image-guided surgery of soft tissue sarcomas to improve the success rates of tumor resections.

**Abstract:**

Surgery is the mainstay of treatment for localized soft tissue sarcomas (STS). The curative treatment highly depends on complete tumor resection, as positive margins are associated with local recurrence (LR) and prognosis. However, determining the tumor margin during surgery is challenging. Real-time tumor-specific imaging can facilitate complete resection by visualizing tumor tissue during surgery. Unfortunately, STS specific tracers are presently not clinically available. In this review, STS-associated cell surface-expressed biomarkers, which are currently already clinically targeted with monoclonal antibodies for therapeutic purposes, are evaluated for their use in near-infrared fluorescence (NIRF) imaging of STS. Clinically targeted biomarkers in STS were extracted from clinical trial registers and a PubMed search was performed. Data on biomarker characteristics, sample size, percentage of biomarker-positive STS samples, pattern of biomarker expression, biomarker internalization features, and previous applications of the biomarker in imaging were extracted. The biomarkers were ranked utilizing a previously described scoring system. Eleven cell surface-expressed biomarkers were identified from which 7 were selected as potential biomarkers for NIRF imaging: TEM1, VEGFR-1, EGFR, VEGFR-2, IGF-1R, PDGFRα, and CD40. Promising biomarkers in common and aggressive STS subtypes are TEM1 for myxofibrosarcoma, TEM1, and PDGFRα for undifferentiated soft tissue sarcoma and EGFR for synovial sarcoma.

## 1. Introduction

Soft tissue sarcomas (STS) are a heterogeneous group of mesenchymal tumors that represent 1% of all malignancies [1]. The incidence in Europe is estimated at 4–5/100,000 per year, accumulating to approximately 18,000 new patients in Europe per year [2,3]. While most STS are diagnosed in the extremities (60%), they can arise anywhere in the body [4]. There are over 50 histological subtypes of STS, each with distinct behavioral, clinical, and prognostic features [5]. Surgery of STS is the mainstay of treatment for localized disease. For the aim of curative surgery, a tumor needs to be removed with a margin of normal tissue as the tumor pseudocapsule and reactive zone are expected to contain tumor cells [6]. Clinical outcome after surgical treatment is highly dependent on surgical resection margins, as tumor-positive margins are clearly associated with local recurrence (LR), and indirectly associated with overall survival [7,8,9,10]. Further, close or positive margins often necessitate the need for adjuvant radiotherapy to reduce the risk for LR with about 50%, but this increases the risk for local complications [11,12]. However, determining the surgical margin is challenging, particularly when tumor tissue is surrounded by vital structures or in STS subtypes with a highly infiltrative growth pattern, such as myxofibrosarcoma (MFS), undifferentiated soft tissue sarcoma (USTS, previously called undifferentiated pleomorphic sarcoma), and synovial sarcoma (SS). In these specific tumors, preoperative surgical planning is complicated by current limitations in preoperative radiological imaging. The infiltrative growth of sarcoma with long slender tails, clearly diagnosed by histology after surgical resection, is sometimes difficult to detect with preoperative imaging [13]. Consequently, despite centralizing STS treatment and (neo)adjuvant treatment modalities, positive margins and LR are still common. Positive margins are 13%, 20% and 28%, with LR rates of 12% (5-year follow up), 40% (10-year follow up), and 45% (5-year follow up) in SS, MFS, and USTS respectively [14,15,16,17,18]. The real-time intraoperative tumor-specific imaging of STS could help the surgeon to discriminate tumor from normal tissue, improving complete tumor resections and reducing LR rates. Near-infrared fluorescence (NIRF) imaging is one of the most upcoming technologies in real-time targeted imaging as it facilitates surgeons to visualize tumor tissue during surgery. It has been explored for various tumor types with promising results and is expected to play an important role in future surgery of STS [19].

Three important parameters define successful NIRF tumor-specific imaging: a tumor-specific biomarker, a targeting moiety conjugated to a fluorescent dye/fluorophore (tracer), and a NIRF camera system. In NIRF imaging, light in the near-infrared (NIR) wavelength is used (650–900 nm). In this region, tissue penetration of light is relatively high, due to low tissue absorption, and the autofluorescence of normal tissue is limited [20]. Light in the NIR region is invisible to the human eye and therefore a dedicated NIRF camera system is needed, which has the advantage that the surgical field is not altered by the fluorescence from the tracer. Clinical NIRF cameras of various companies are available [21].

The search for a tumor-specific biomarker for NIRF imaging of STS is complex, because of the rarity and heterogeneity of the disease. The ideal biomarker should be highly and homogenously expressed on tumor cells of most subtypes of STS, while being absent on adjacent healthy tissue. Like for other cancers, the biomarker should preferably be located on the cell surface of malignant cells to permit direct targeting and have the possibility of internalization (endocytosis of an extracellular molecule upon binding to a specific protein on the cell surface) to facilitate a long-lasting fluorescence signal. Ideally, this biomarker is still present on residual cells after neoadjuvant therapy.

Fluorescent tracers for tumor biomarkers are generated by the conjugation of a fluorescent dye/fluorophore to a targeting moiety. Various fluorophores are available and some are clinically approved [22]. Targeting moieties consist of proteins, like monoclonal antibodies or fragments thereof, peptides, RNA aptamers, or other small synthetic molecules. Monoclonal antibodies are the most widely used targeting moieties in biotherapy and imaging. The advantages of antibodies are their specificity, affinity, flexibility, and relatively long plasma half-life. To minimize immune reactions, human(ized) versions are mostly used. A disadvantage of antibodies for the use of imaging is the relatively high costs of development, which is particularly relevant for rare diseases like STS. In the past decade, therapeutic antibodies have been equipped with NIRF dyes and evaluated for imaging of common cancer types, like breast and colorectal cancer [19].

Elaborating on this approach, the aim of this systematic review is to select candidate biomarkers for specific intraoperative NIRF imaging of soft tissue sarcomas. STS are a rare and heterogeneous group of tumors. The development of a specific tracer for NIRF imaging that is not already clinically used in therapy would be very challenging as it would be costly and time consuming. Therefore, the search is restricted to clinically available monoclonal antibodies of which the safety profiles are already demonstrated and a translation towards a tracer for NIRF imaging can be expected. The overall purpose of this evaluation is to find optimal biomarkers for the three most common and aggressive STS subtypes MFS, USTS, and SS, which account for challenging resections and currently result in high rates of local recurrences.

## 2. Materials and Methods

### 2.1. Search Strategy

An initial search was performed to find clinically available monoclonal antibodies targeting STS. The EU Clinical Trials Register (www.clinicaltrialsregister.eu/) and clinical trials.gov (clinicaltrials.gov/) databases were searched with the keyword “Soft tissue sarcoma”, and all clinically available monoclonal antibodies targeting STS were listed. Next, a PubMed search with the respective biomarkers targeted by those monoclonal antibodies was created with the assistance of a medical librarian (Appendix A). The search was done in August 2019 and updated in September 2020 due to the publication of multiple relevant articles between August 2019 and September 2020. This systematic review was performed following the Preferred Reporting Items for Systematic Reviews and Meta-Analysis (PRISMA) guidelines of 2009 (registration ID: CRD42020206473) [23].

### 2.2. Eligibility Criteria

Studies were eligible for inclusion if they met the following criteria: (1) report of expression of cell surface-expressed biomarkers in STS for which a clinically available antibody was present, (2) at least 95% of the included tumor samples were primary STS, (3) sample size of at least 4, (4) published in the English language, and (5) full text was available. The eligibility of the studies was assessed by two authors (Z.R. and A.N.S.). Disagreements were discussed with a third reviewer (P.B.A.A.v.D.). Animal studies, xenograft studies, cell line studies, articles without positive and negative control samples, case reports, reviews, viewpoints, conference reports, meeting abstracts, letters to journals, or editors were excluded.

### 2.3. Data Extraction

The following data were extracted from eligible studies: target characteristics, sample size, type of sample, percentage of positive STS samples, localization of expression, pattern of expression, positive and negative controls, internalization, and previously imaged. A second tumor type-independent search was performed for data on internalization and previous imaging of targets where no information was found after the first search (Appendix B). Data on safety profiles of monoclonal antibodies was acquired through the search of Appendix A.

### 2.4. Biomarker Selection Scoring System

In order to select the optimal biomarkers for tumor specific NIRF imaging in STS, we developed a target selection scoring system. The scoring system is based on the modified version of the Target Selection Criteria (TASC), developed by Bosma et al. [24]. The scoring system is based on five domains (see Table 1).
Sample size. The number of samples indicate how much evidence is acquired.Percentage of biomarker-positive STS samples. This is calculated based on the amount of STS samples that positively showed presence of the biomarker in each included article, independent of the percentage of positive tumor cells within each sample. Immunohistochemistry was used to assess the percentage of positive STS in tissue samples.Pattern of expression. Ideally, the target is expressed diffusely by all tumor cells (particularly at the tumor border) to guide surgical resection. The pattern of expression is defined as diffuse when expression is randomly spread throughout the tumor sample and focal when expression is located in a specific region of the tumor sample. When different samples show variable expression patterns (diffuse and focal), the expression pattern for the whole cohort is defined as heterogeneous. No distinction was made based on exact location of expression within tumor samples. While this review included studies evaluating tissue samples and tissue microarrays, data regarding the pattern of expression was extracted from studies including tissue samples.Internalization. This is important because internalization after binding of the tracer creates a long-lasting signal for tumor-specific imaging.Previously imaged. If there is prove that imaging is possible, it has more potential to be translated to the clinics. The distinction between imaging with or without NIRF is important for its applicability in NIRF imaging. This criterium was tumor type independent.

The maximum score for a target is 9 points, 7 was chosen as the cut-off value for promising targets for tumor specific NIRF imaging in STS.

## 3. Results

### 3.1. Study Selection

Our analysis of the EU Clinical Trials Register (https://www.clinicaltrialsregister.eu/) and clinical trials.gov (https://clinicaltrials.gov/) revealed the following clinically available monoclonal antibodies targeting STS-associated cell surface-expressed biomarkers (Table 2): Ontuxizumab (MORAb-004) [trial number: NCT01574716] targeting tumor endothelial marker 1 (TEM1), recombinant monoclonal antibody Aflibercept [NCT00390234], and humanized monoclonal antibody Bevacizumab [NCT03913806] targeting vascular endothelial growth factor A (VEGF-A), thereby indirectly targeting vascular endothelial growth factor receptor-1 (VEGFR-1) and vascular endothelial growth factor receptor-2 (VEGFR-2), Ramucirumab [NCT04145700] targeting VEGFR-2, Cetuximab [NCT00148109] targeting epidermal growth factor receptor (EGFR), Ganitumab (AMG 479) [NCT03041701], Teprotumumab [NCT00642941], Cixutumumab [NCT01016015] and Figitumumab [NCT00927966] targeting insulin-like growth factor 1 receptor (IGF-1R), Olaratumab [NCT03126591] targeting platelet derived growth factor α (PDGFRα), APX005M [NCT03719430] targeting cluster of differentiation 40 (CD40), Atezolizumab [NCT03474094], Avelumab [NCT04242238], Durvalumab [NCT03317457], and Envafolimab [NCT04480502] targeting programmed death-ligand 1 (PD-L1), ABBV-085 [NCT02565758] targeting leucine-rich repeat containing 15 (LRRC15), CAB-ROR2-ADC [NCT03504488] targeting receptor tyrosine kinase-like orphan receptor 2 (ROR2) and Ipilimumab [NCT04118166], and Tremelimumab [NCT03317457] targeting cytotoxic T-Lymphocyte-associated protein 4 (CTLA-4).

The PubMed search based on the cell surface-expressed biomarkers targeted by clinically available monoclonal antibodies identified 1856 articles (Figure 1). After screening the titles and abstracts, 1604 articles were excluded. Subsequently, 252 full-text articles were assessed for eligibility, of which 171 articles did not meet eligibility criteria; 107 articles did not study expression of the included biomarkers on human STS cells, for 19 articles data was not suitable for extraction, 16 articles had a sample size of less than 4 samples, 11 articles did not have full-text available, 10 articles had more than 5% of samples which were not primary STS and therefore their results were no longer a valid representation of STS samples, and 8 articles were reviews or letters to journals without an accompanying methods section. Data regarding internalization and previously imaged was not always described in STS. Therefore, a separate search was performed to obtain these data from other tissue types (Appendix B). This resulted in an additional 16 included articles. Ultimately, 97 articles were included for this review.

### 3.2. Candidate Biomarkers

A modified Target Selection Criteria TASC)-scoring system was applied to eleven cell surface-expressed biomarkers (Table 1). Seven promising candidate targets for NIRF imaging emerged with a minimum score of 7 out of 9. The biomarkers arranged in descending order based on their scores were: TEM1 (9), VEGFR-1 (8), EGFR (8), VEGFR-2 (7), IGF-1R (7), PDGFRα (7) and CD40 (7). Further details of these biomarkers are described below and in Table 2, focusing on their physiological role, expression in STS, the availability of clinically used monoclonal antibodies targeting these biomarkers, and latest developments.

#### 3.2.1. TEM1

Tumor Endothelial Marker 1, also referred to as Endosialin or CD248, is a highly glycosylated type I transmembrane protein classified among the C-type lectin-like domain superfamily 14. It has been suggested that TEM1 plays a critical role in wound healing and angiogenesis [121,122]. Moreover, while it is expressed minimally in normal conditions, it is markedly upregulated in the setting of injury and malignant tumor growth. In (soft tissue) sarcomas TEM1 was observed to be present on malignant cells [28]. Stromal TEM1 promotes spontaneous metastasis and TEM1-expressing pericytes were shown to facilitate distant site metastasis by stimulating tumor cell intravasation [123]. Furthermore, TEM1 expression is associated with enhanced tumor growth, presumably due to tumor-specific angiogenesis [124].

The presence of the biomarker in STS samples, regardless of the percentage of positive tumor cells, was determined on both tumor and stromal cells for TEM1. In STS, 77% (range 55–100%, *n* = 768) of the samples showed presence of TEM1 on average, reported in 4 different articles [26,27,28,29]. Staining was performed in 17 subtypes of STS (Appendix C). The expression pattern for TEM1 was diffuse. Corresponding to the expression in other cancer types, TEM1 expression is correlated with advanced tumor grade in STS [121,125].

In MFS it was demonstrated that TEM1 was present in all 34 investigated samples, with a diffuse pattern of expression [27]. Staining was negative or very limited in normal adjacent tissue such as muscular fascia and peripheral nerve bundles. For USTS an average of 81% (range 73–89%, *n* = 128) of the samples expressed TEM1, with a diffuse pattern of expression [28,29]. In SS, 71% (range 62–80%, *n* = 70) of the tissue samples stained positive for TEM1. The pattern of expression was heterogeneous with samples expressing TEM1 either focally or diffusely. Besides, Thway et al. [29] demonstrated in representative images that the spindle cell component of biphasic SS samples is positive, while the glandular epithelial areas are negative. Regarding monophasic SS, both positive and negative samples were reported [28,29]. Data are summarized in Table 3.

Exclusively Ontuxizumab has been clinically investigated as a therapeutic drug in STS [130]. However, it still needs to be modified into a NIRF imaging tracer. A high-affinity human single-chain variable fragment (scFv)-Fc fusion protein (78Fc) targeting TEM1 has been engineered and conjugated with the near-infrared fluorochrome VivoTag-S750, which proved to be an efficient tracer in preclinical osteosarcoma and lung cancer models [25,27,124,127].

In conclusion, TEM1 can be targeted in NIRF imaging by Ontuxizumab upon conjugation to a NIRF dye and small proteins have been produced pre-clinically for similar purposes. A major advantage of TEM1 is that it has minimal to no expression on adjacent normal tissue and therefore it is characterized by a high tumor-to-background ratio. Additional benefits are its diffuse pattern of expression, the high frequency of positivity (STS 77%, MFS 100%, USTS 81% and SS 71%), and its correlation with advanced tumor grades. A disadvantage is its heterogeneous pattern of expression in the SS subtype with samples illustrating focal expression of TEM1.

#### 3.2.2. VEGFR

The VEGFR family consists of the 3 members VEGFR-1, -2 and -3 which are receptors for ligands VEGF-A, -B, -C, -D, and Placenta Growth Factor [22]. The receptors contain a split tyrosine kinase domain and a ligand-binding part. The individual VEGFR members have separate roles in various signaling pathways, but as a family they collectively function as the principal driver of angiogenesis and lymph angiogenesis. Hence, VEGFRs are mainly expressed on vascular and lymphatic endothelial cells in healthy tissue [21,130,131]. In various tumor types, including STS, they are expressed by both endothelial cells and tumor cells [131]. Here they stimulate tumor growth [132]. VEGFR-1 and VEGFR-2 have been clinically targeted by antibodies in STS, in contrast to VEGFR-3. Therefore, only VEGFR-1 and VEGFR-2 will be evaluated.

VEGFR-1 presence was found in an average of 76% (range 22–100%, *n* = 477) of the STS patients in 8 different studies [32,33,34,35,36,37,38,39]. Staining was performed in 15 STS subtypes (Appendix C). The VEGFR-1 expression pattern was demonstrated to be diffuse. Expression was found in the cytoplasm, and on the nuclear and cell membrane [32,35]. VEGFR-2 expression was present in 71% (range 11–100%, *n* = 449) on average in 9 different studies, and 16 STS subtypes were evaluated [33,34,35,36,38,39,79,80,81]. The pattern of expression was heterogeneous, and expression was found in the cytoplasm, and on the nuclear and cell membrane [35,81]. Interestingly, Kilvaer et al. [131] states that VEGFR overexpression is correlated with an increased tumor grade.

No data were found for VEGFR immunohistochemical staining in MFS. One paper reported on the presence of VEGFR-1 and VEGFR-2 in USTS and SS [36]. VEGFR-1 and VEGFR-2 expression was found in 68% and 6% of 81 USTS samples, respectively. In SS, this was 70% and 4% for respectively VEGFR-1 and VEGFR-2 in 27 samples (Table 3). Moreover, the pattern of expression was described for neither USTS nor SS [36]. Additionally, no distinction was made between monophasic and biphasic SS in the published data.

Ramucirumab binds to VEGFR-2 and is currently in its recruitment phase for clinical testing in SS [133]. Besides, VEGFR-1 and VEGFR-2 may be targeted indirectly using Bevacizumab-IRDye800CW or Aflibercept upon conjugation to a NIRF dye [29,30,134,135]. Recently published study results showed visualization of all 15 included STS patients with Bevacizumab-IRDye800CW targeting VEGF-A. In this paper, in vivo tumor-to-background ratios of 2.0-2.5 were found with doses of 10-25mg tracer and no tracer-related adverse events occurred within 2 weeks after surgery [136]. Additionally, targeting tumors with Bevacizumab-IRDye800CW has been investigated extensively in clinical trials for several tumor types [134,135,136,137]. Here, its tolerable safety profile was confirmed in primary breast cancer patients [138].

In conclusion, VEGFR-1 and VEGFR-2 are receptors that may be targeted indirectly with a tracer, Bevacizumab-IRDye800CW, that has already widely proven its benefit in multiple cancer types. The direct targeting of VEGFR-2, however, may additionally be performed with Ramucirumab. Major advantages of VEGFR-1 are the high frequency of positivity in STS (76%), the diffuse pattern of expression in tumors and increasing expression associated with enhanced tumor grade. However, while VEGFR-1 is commonly present in USTS and SS, there is no data concerning its pattern of expression in these STS subtypes. Furthermore, advantages of VEGFR-2 are its high presence of 71% in STS samples and increasing expression associated with enhanced tumor grade. Disadvantages are a heterogeneous, and therefore unpredictable, pattern of expression in the evaluated tissue samples and the fact that only 6% of USTS and 4% of SS are positive. Additionally, both VEGFRs are commonly expressed in healthy tissue, potentially resulting in a low tumor-to-background ratio.

#### 3.2.3. EGFR

Epidermal Growth Factor Receptor is a transmembrane glycoprotein belonging to the ErbB/HER family together with 3 additional distinct receptor tyrosine kinases: ErbB2/HER2, ErbB3/HER3, and ErbB4/HER4 [139]. Seven different ligands trigger intracellular signals for fundamental cellular functions including proliferation, differentiation, migration and survival of tumor cells [140,141]. EGFR is mainly expressed in proliferating keratinocytes [142,143]. In tumors, EGFR overexpression can trigger tumor invasion and metastasis. Furthermore, it is a central regulator of autophagy, which is strongly involved in resistance to cancer therapies [144,145].

EGFR expression in STS was described in 36 scientific papers [27,42,43,44,45,46,47,48,49,50,51,52,53,54,55,56,57,58,59,60,61,62,63,64,65,66,67,68,69,70,71,72,73,74,75,76]. The presence of EGFR on STS tissue was observed in an average of 53% of the samples (range 0–100%, *n* = 1918). Expression was evaluated in 29 different subtypes of STS (Appendix C). The pattern of expression was diffuse. Importantly, EGFR expression in STS was strongly correlated to higher histological grade [46,48,70].

In MFS, EGFR presence was observed in an average of 38% (range 0–89%, *n* = 97) of the samples in 3 articles (Table 3) [26,53,65]. This wide range might be explained by the fact that 1 article included 10 low-grade MFS samples of which none expressed EGFR. The remaining 2 articles had a higher percentage of positive samples with a diffuse pattern of expression. This confirms the positive correlation of EGFR expression with increased histological grade STS [26,53,65]. For USTS, EGFR expression was detected in an average of 62% (range 5–95%, *n* = 287) of the samples with a heterogeneous pattern of expression. Similar to MFS, a wide range was observed with 1 article reporting 5% of 200 samples to be positive for EGFR staining, 1 article reporting 58% in 24 samples, and 2 articles reporting 91% and 95% positive samples in 44 and 19 samples, respectively. Here, the correlation to increased histological grade could not explain the variable expression [50,57,65,70]. Lastly, EGFR presence was seen in an average of 86% (range 71–100%, *n* = 160) of the SS samples. The pattern of expression was noticeably heterogeneous, extending from focal to diffuse expression [52,58,66,69,70,71]. Furthermore, Gusterson et al. [58] and Sato et al. [66] compared the spindle cell and epithelial components of biphasic SS samples. They described that the former is strongly positive, whereas the latter is mainly negative for EGFR expression. Regarding monophasic SS, both positive and negative samples were reported.

Currently, Cetuximab is the only clinically investigated EGFR-targeting monoclonal antibody for STS [146]. It has been conjugated to IRDye800 and examined in several clinical trials in other tumor types. To appraise its utility in the detection of metastatic lymph nodes in pancreatic cancer, a total of 144 human lymph nodes were evaluated *ex-vivo*. The Cetuximab-IRDye800 conjugate demonstrated a sensitivity and specificity of 100% and 78% [147]. Additionally, no grade 2 or higher adverse events were observed with Cetuximab-IRDye800 in glioblastoma and head and neck squamous cell carcinoma [148,149].

A clinical trial investigating the use of ABY-029, an affibody conjugated to IRDye800CW targeting EGFR, is in the recruitment phase for targeting STS [150]. Based on pre-clinical research it is a promising tracer for STS and is safe for human use [41,151]. Other clinical trials in their recruitment phase explore the use of Panitumumab-IRDye800 in imaging of head and neck cancer, lung cancer, and metastatic lymph nodes [152,153,154].

In summary, there are multiple promising tracers available which can be applied for NIR fluorescence-guided surgery in STS. Main advantages of EGFR, apart from the readily available tracers, are its diffuse pattern of expression in STS in general, the increased expression in STS of higher histological grade, and the high frequency of expression (88%) among SS samples. Yet, some drawbacks are the mediocre percentage (54%) of positive tumor samples in STS in general and the highly heterogeneous expression pattern in SS.

#### 3.2.4. IGF-1R

Insulin-like Growth Factor 1 Receptor is a receptor tyrosine kinase that is activated upon binding with IGF-1 or IGF-2. Under normal physiological circumstances, this provokes a chain of signaling events that induce cellular transformations such as hypertrophy in skeletal muscle. IGF-1R is upregulated in multiple malignancies, including prostate, breast and lung cancer, where it is involved in tumor growth. Besides, it enables cancer cells to resist the cytotoxic properties of radiotherapy and chemotherapeutic drugs by inducing an anti-apoptotic effect [24].

IGF-1R presence was detected in 63% (range 25–100%, *n* = 507) of STS samples on average in 9 different studies [63,64,82,90,126,128,129,155,156]. Staining was performed in 15 subtypes of STS (Appendix C). The receptor was dispersed diffusely in the cytoplasm, and on the nuclear and cell membrane [62,82,128]. No correlation between histological grade and IGF-1R expression was observed [64,128].

No data are available on IGF-1R presence in MFS. Presence of IGF-1R in USTS and SS was evaluated in 1 and 4 articles respectively [81,82,126,128,129]. IGF-1R presence was found in 25% of the USTS samples (*n* = 120), while in SS an average of 57% (range 35–100%, *n* = 195) of the samples stained positive. The pattern of expression was described for neither (Table 3). However, Friedrichs et al. [155] reported that vast areas of tumorous tissue showed membranous staining in monophasic (comprising spindle cells) SS. In contrast, biphasic SS samples displayed predominantly positive staining in the epithelial component. Regarding monophasic SS, both positive and negative samples were reported [81,126,128,129].

Clinical trials targeting IGF-1R in STS have been conducted with Teprotumumab, Cixutumumab, Figitumumab, and Ganitumab [157,158,159,160,161]. Nevertheless, these monoclonal antibodies have not been evaluated for their potential in NIRF imaging.

AVE-1642, a humanized anti-IGF-1R antibody, labelled with Alexa 680 has been pre-clinically investigated in in vivo breast cancer models and adequately identified receptor expression [162].

Overall, IGF-1R may be targeted in NIRF imaging by several potential antibodies after conjugation to a NIRF dye. In addition, pre-clinical advances have resulted in promising tracers that may find future clinical use. An advantage of IGF-1R is its relatively common (63%) presence in all STS samples. However, its expression has no correlation with tumor grade, and data on pattern of expression in MFS, USTS and SS is limited.

#### 3.2.5. PDGFR

Platelet-Derived Growth Factor is a receptor tyrosine kinase characterized by two isoforms, PDGFRα and PDGFRβ [163]. The receptors can be activated after binding by ligands from the PDGF-family. Upon activation, PDGFR is known to control angiogenesis in endothelial cells, and cell migration and growth in mesenchymal cells. Moreover, in healthy tissue both PDGFRs are mainly expressed in mesenchymal cells during inflammation, whereas during non-inflammatory conditions the expression is minimal [164,165]. In tumor biology, PDGFR activation stimulates cell growth and enhances metastatic behavior by attracting fibroblasts, which secrete factors that promote proliferation and migration of tumor cells. Both PDGFRα and -β are expressed by tumor cells of STS, yet expression of specifically PDGFRα is evaluated in this review as a monoclonal antibody against this receptor has been clinically tested in STS, while not against PDGFRβ [42,46,166,167].

Based on the literature search, PDGFRα was present in 64% of STS samples on average (range 0–100%, *n* = 1536) in 21 different articles [27,34,36,38,42,43,44,45,46,47,48,49,51,82,86,87,88,89,90,91,92]. Expression was evaluated in 22 different subtypes of STS (Appendix C). The pattern of expression was diffuse, and expression was identified in the cytoplasm, and on the nuclear and cell membrane of the tumor cells [45,86,168].

PDGFRα expression in the specific STS subtypes of interest, MFS, USTS, and SS, were evaluated separately in 1, 4, and 5 articles, respectively. In MFS PDGFRα was present in 77% of 34 tissue samples [27]. In USTS, 78% of the tumors (range 63–99%, *n* = 475) were positive for PDGFRα, while for SS 69% (range 44–84%, *n* = 136) stained positive. Moreover, expression was reported to be diffuse in USTS. No data regarding the pattern of expression of MFS and SS were reported [35,50,88,91,126,155]. However, opposing data was published regarding differences in expression of either spindle cell or epithelial components in biphasic SS. While Fleuren et al. [89] displayed images where exclusively the spindle cell component expressed PDGFRα, Lopez-Guerrero et al. [92] reported that membranous staining was more prominent in the epithelial component. Regarding monophasic SS, both positive and negative samples were reported. Data are summarized in Table 3.

Multiple drugs targeting PDGFRα are currently FDA approved or subject to clinical trials. However, Olaratumab is the only monoclonal antibody that has been clinically investigated for STS. It binds specifically PDGFRα [169]. No clinical NIRF imaging studies have been performed using Olaratumab conjugated with a fluorophore in any cancer type.

In summary, PDGFRα may be targeted in NIRF imaging by Olaratumab after conjugation to a NIRF dye. The advantages of PDGFRα are its relatively regular (65%) presence in STS samples and its diffuse pattern of expression in specifically USTS with 78% of samples expressing PDGFRα. The disadvantages are the non-reported patterns of expression for MFS and SS, and no article addressed a correlation between enhanced PDGFRα expression and histological grade.

#### 3.2.6. CD40

Cluster of Differentiation 40 is a member of the tumor necrosis factor family and can be ligated by CD40 Ligand (CD40L). CD40 is detected on dendritic cells, B-cells and myeloid cells that can mediate cytotoxic T-cell priming upon CD40L ligation [170]. Moreover, it is constitutively expressed on platelets, smooth muscle cells, and endothelial cells [166]. In cancer, CD40 has been found in nearly all B-cell malignancies and many solid tumors, where it induces a direct cytotoxic effect in the absence of immune accessory cells [167]. It is hypothesized that it confers a growth and survival stimulus via signaling pathways such as PI3Kinase/Akt and NFκB and/or that it modulates anti-tumor immune responses [168].

CD40 was present in 62% of STS samples (range 17–86%, *n* = 153) on average in 4 different scientific papers [95,96,97,98]. The pattern of expression was diffuse, when assessed in 7 subtypes (Appendix C). Expression was observed on the membrane and in the cytoplasm of tumor cells [95,96,97,98]. No association between enhanced CD40 expression and histological grade was found after comparing low-grade to high-grade STS samples [97]. Furthermore, no articles published data regarding CD40 expression on MFS, USTS and SS separately.

A phase II clinical trial applying APX005M, a second-generation agonistic CD40 monoclonal antibody, combined with Doxorubicin in STS is currently recruiting participants [171]. Nonetheless, the antibody has not yet been evaluated for NIRF imaging and no other CD40-targeting drug has thus far been clinically examined for CD40.

Apart from 2 articles focusing on respectively B-cell activation by targeting CD40 with nanoparticles and cerebral ischemia by targeting CD40 with an anti-CD40 antibody conjugated to Cy5.5, no pre-clinical advances in the field of NIRF imaging can be addressed using CD40 as a target [94,172].

In conclusion, APX005M may be utilized as tracer after conjugation to a NIRF dye for imaging in STS. Pre-clinical studies have developed tracers targeting CD40, yet these have not been tested in STS models thus far. Advantages of CD40 are a diffuse pattern of expression and the fact that expression is relatively common (62%) in STS samples in general. Disadvantages are the small number of evaluated STS samples and the lack of data regarding CD40 expression in MFS, USTS and SS.

### 3.3. Potential NIRF Imaging Tracers Safety Profile

In this review, 7 potential targets for fluorescence-guided surgery of STS (TEM1, VEGFR-1, EGFR, VEGFR-2, IGF-1R, PDGFRα, and CD40) were selected based on antibodies that are clinically available and mostly used in the antibody-based therapy of STS. Several tracers have already proven to be well suitable for NIRF imaging. Among these tracers, Bevacizumab-IRDye800CW targeting VEGF-A (indirectly VEGFR-1 and VEGFR-2) has already shown promising results in STS [136]. Besides, Cetuximab-IRDye800 targeting EGFR is an adequate tracer in several tumor types [147,148,149]. This section elaborates on clinically available monoclonal antibodies which can be modified into tracers: Ontuxizumab targeting TEM1, Teprotumumab, Cixutumumab and Figitumumab targeting IGF-1R, and Olaratumab targeting PDGFRα [130,159,160,161,173,174,175,176]. APX005M targeting CD40 is currently under investigation and therefore its efficacy and safety profile in STS are yet to be determined. In contrast to therapy, a single dose of tracer is injected for imaging and an increase in adverse effects compared to therapy is not expected. Further, no increase in adverse effects is expected after conjugation of a fluorophore and antibody [136,177,178,179]. This paragraph summarizes the safety profiles of each clinically available monoclonal antibody extracted from advanced clinical trials conducted with STS-patients to evaluate their potential for translation towards NIRF imaging. Only high grade (grade ≥ 3) Adverse Events (AE) are displayed.

Ontuxizumab was compared to a placebo when both were combined with Gemcitabine and Docetaxel. While the total of grade ≥3 AEs was not reported, the incidence of Serious Adverse Events (SAE) was comparable between Ontuxizumab and placebo (50% vs. 48%). The most frequent treatment related SAEs were pyrexia (4% vs. 0%) and anemia (1% vs. 3%) (Appendix D). No substantial differences were observed in laboratory values or electrocardiogram parameters [130].

Targeting IGF-1R, Teprotumumab, Cixutumab, and Figitumumab were investigated as a monotherapy. These trials have reported a minor incidence of high-grade AEs. AEs such as hyperglycemia, pain, thrombocytopenia, and vomiting were the most common high-grade AEs with incidences ranging from 3–5%. Of all included study subjects, 10% and 17% of patients acquired grade ≥3 AEs for Teprotumumab and Figitumumab, respectively. Among these 3 antibodies, Teprotumumab was demonstrated to have the most tolerable and Cixutumumab the most toxic safety profile in STS [159,160,161].

Two studies on Olaratumab reported grade ≥3 Adverse Events (AE) in 58–67% of the patients when combined with Doxorubicin alone [174,180]. In addition, 2 studies observed contrasting AEs when Olaratumab plus Doxorubicin was compared to Doxorubicin. A phase 2 trial observed an increased incidence of high-grade AEs for the combination therapy while a phase 3 trial found no significant differences and therefore concluded no additional adverse events to be attributed to Olaratumab [179,180,181]. Hematologic grade ≥3 AEs were most common in these trials with incidences reaching 40–50% (Appendix D).

## 4. Discussion

### 4.1. Research Aim

The success of surgical treatment for localized STS highly depends on complete tumor resection as positive margins are associated with LR and decreased overall survival. Determining the surgical margin is a major challenge for STS surgeons as they generally try to balance the aim of a functional limb against the risk of LR. Real-time tumor-specific imaging can improve surgical margins by visualizing tumor tissue during resection. This review selected TEM1 (score 9), VEGFR-1 (score 8), EGFR (score 8), VEGFR-2 (score 7), IGF-1R (score 7), PDGFRα (score 7), and CD40 (score 7) as the most promising cell surface-expressed biomarkers for tumor-specific NIRF imaging in STS, for which clinically available monoclonal antibodies are already present. Additionally, these potential future NIRF tracers, which are antibodies that have already been clinically tested in STS but not yet conjugated to a NIRF-dye for imaging practices, are expected to be safe for their use in NIRF guided surgery.

### 4.2. Comparing the Selected Biomarkers

All the suitable biomarkers have already been evaluated for NIRF imaging pre-clinically, demonstrating their potential [25,29,30,77,82,84,93]. Furthermore, all the selected cell surface-expressed biomarkers internalize after binding with an antibody (derivative) [24,29,39,76,81,92,156]. This causes a better tumor-to-background ratio and a long-lasting signal important for fluorescence-guided surgery [19,20]. However, the indirect targeting of VEGFR-1 and VEGFR-2 by targeting VEGF-A with, for instance, Bevacizumab-IRDye800CW, has not been proven to result in internalization of tracers.

TEM1 and VEGFR-1 were most frequently present in STS samples, 77% and 76% respectively. VEGFR-2 was third most frequently expressed (71%), followed by PDGFRα (64%), IGF-1R (63%), CD40 (62%), and EGFR (53%). Furthermore, apart from CD40 (*n* = 153), presence of every biomarker of the top 7 has been studied in a large number of STS samples. Therefore, the summarized data in this review are a good representation of biomarker presence in STS patients: EGFR (*n* = 1918), PDGFRα (*n* = 1536), TEM1 (*n* = 768), IGF-1R (*n* = 507), VEGFR-1 (*n* = 477), and VEGFR-2 (*n* = 449).

A particularly important parameter for successful NIRF imaging, which is not included in the TASC score, is the tumor-to-background ratio of a biomarker. With the currently available literature it is impossible to address the expression of each biomarker in healthy tissue, and thus the tumor-to-background ratio, because data on the expression of the biomarkers in normal tissue is very limited. Nevertheless, VEGFR-1 and VEGFR-2 are highly expressed in healthy tissue, while TEM1 and PDGFRα are biomarkers with low expression in healthy tissue. TEM1 has already shown high tumor-to-background ratios with immunohistochemistry [27]. However, both biomarkers are expressed in inflammatory tissue as well as in tumors [28,182]. As STS can be surrounded by inflammation during their growth, it is possible that no clear distinction can be made between tumor and surrounding inflammatory tissue [183]. Unfortunately, none of the selected studies reported on inflammation status of surrounding tissue. In addition, neoadjuvant therapy is frequently used in STS treatment. Successful fluorescence guided surgery is only possible if the overexpression of cell surface-expressed biomarkers is preserved after neoadjuvant therapy. It was demonstrated that EGFR, TEM1, and PDGFRα expression is preserved after neoadjuvant radiotherapy of MFS [27]. This has also been confirmed for EGFR in SS [127]. No other data is available on the expression of these or the remaining evaluated biomarkers after neoadjuvant therapy in STS. Therefore, further research is needed to assess if surrounding inflammatory tissue or neoadjuvant therapy interferes with tumor border identification in STS.

### 4.3. MFS, USTS and SS

We chose to focus on MFS, USTS, and SS because of their aggressive and infiltrative growth pattern. TEM1 was present in 100% of the MFS samples (Table 3). Besides, its pattern of expression was diffuse in all tested MFS samples [27]. This indicates that TEM1 is likely to be extensively expressed in tumors of every individual MFS patient. Besides, a sharp contrast between tumor and adjacent normal tissue, such as fascia, muscle, and fat, was seen on microscopic pictures of stained MFS samples. This clearly identifies the tumor border and therefore TEM1 seems the most promising biomarker to facilitate complete MFS resections using NIRF imaging [27].

For USTS the average presence of TEM1 and PDGFRα was 81 and 79% of the tumor samples. Apart from being expressed in a substantial percentage of USTS samples, TEM1 and PDGFRα were primarily expressed diffusely [27,28,35,50,126]. However, there is no data published regarding contrast between expression on tumor and normal tissue in USTS. According to the human protein atlas TEM1 and PDGFRα expression is not detected in skeletal muscle tissue and adipose tissue. For smooth muscle tissue, TEM1 displays low expression, while PDGFRα is not detected [184,185]. These characteristics suggest that TEM1 and PDGFRα are promising biomarkers for NIRF imaging in USTS patients.

In SS, the presence of TEM1 and EGFR was demonstrated in 71% and 86% of the assessed samples, respectively. EGFR and TEM1 are both characterized by a variable expression pattern in SS [28,29,52,58,66,69,70,71]. Moreover, both targets are reported to be not or minimally expressed in the epithelial components of biphasic SS tumors, while it was expressed in the spindle cell components. This might complicate NIRF imaging of biphasic SS tumors when solely targeting either of these biomarkers. Interestingly, EGFR remains present on SS after neoadjuvant radiotherapy. This has not been researched for TEM1, therefore providing EGFR a further advantage over TEM1 [127].

Lastly, most biomarkers are not present in 100% of the evaluated STS (subtype) tumor samples. The disadvantage of not knowing expression in advance to surgery can be overcome by evaluating the expression of each biomarker in preoperative biopsies to assess which biomarker would be most appropriate to target for NIRF imaging during surgery.

### 4.4. Comparison of Potential NIRF Imaging Tracers

Several monoclonal antibodies targeting STS have already been adjusted to tracers suitable for NIRF imaging and additional monoclonal antibodies used in therapy may be applicable for future NIRF imaging in STS after conjugation to a fluorescent dye/fluorophore. Five distinct antibodies have been assessed for their toxicity profile in STS (Appendix D). Nevertheless, comparing the results of these drugs is complicated, since Olaratumab and Ontuxizumab have solely been investigated combined with chemotherapeutic agents. Still, no evident increase in high-grade toxicity was detected for either antibodies when compared to placebo suggesting a tolerable safety profile. These results are confirmed in trials investigating Olaratumab in metastatic gastrointestinal stromal tumor (GIST) and Ontuxizumab in metastatic colorectal cancer where respectively 10 and 11% grade of ≥3 treatment-related adverse events were reported [175,186]. These data are similar to the percentages of patients acquiring grade ≥3 AE after treatment with IGF-1R targeting antibodies (Teprotomumab, Figitumumab and Cixutumumab) and therefore all antibodies studied here can be safely modified into NIRF imaging tracers.

It should, however, be emphasized that data on toxicity in antibody-based therapy are presumably an overestimation for imaging, because doses of antibodies injected for NIRF imaging are substantially lower compared to therapeutic doses. For instance, a single dose of 10mg Bevacizumab-IRDye800CW was found to be optimal for NIRF imaging in STS, whereas therapeutic doses comprise of 5–15mg/kg Bevacizumab every 2–3 weeks [134,185,186]. Consequently, the serum concentration of the antibody (conjugated to a fluorophore) is lower when used for NIRF imaging and less toxicity of these monoclonal antibodies is expected [181]. Preferably, dose-finding studies, where single and low doses of the five evaluated compounds have been given to STS patients, should be reviewed to predict toxicity when used for NIRF imaging, yet such articles have not been published.

### 4.5. Strengths and Limitations

The first limitation is that the heterogeneity of the included studies complicates ranking of the biomarkers. Studies have used various antibodies for immunohistochemistry. The percentage of positive tumors may be variable depending on type of antibodies, dilutions, epitope, and clone used [187]. Also, immunohistochemistry protocols differ between labs which may cause variable results while the same type of antibodies is used. This creates discrepancy in immunohistochemical results published by different researchers. Secondly, the heterogeneity of STS complicates selecting the optimal biomarkers. There are over 50 subtypes of STS, and different subtypes have different biomarker expression patterns [71]. Therefore, finding one optimal biomarker for each subtype is challenging.

A strength of this study is our focus on MFS, USTS, and SS as they are STS subtypes which show an infiltrative growth pattern, and consequently have high percentages of positive margins and high percentages of LR. Patients with these subtypes might benefit the most from implementation of NIRF imaging. Nevertheless, published data regarding some biomarkers in MFS is scarce. Another strength is that clinically available monoclonal antibodies were the starting point of this systematic review. This was because primary development of a NIRF tumor-specific tracer for a rare disease such as STS is time consuming and costly which hampers rapid clinical implementation. However, alternative antigens that might be interesting for tumor-specific imaging in STS can be missed because no clinically available antibodies (or antibody derivatives) are available. Nevertheless, clinical implementation is of utmost importance to prove feasibility of NIRF imaging for STS surgery and subsequently stimulate primary development of STS specific tracers. This progression is enabled by this review as each evaluated biomarker is accompanied by a clinically available antibody (derivative) that can be transformed into a NIRF tracer.

## 5. Conclusions

In STS, TEM1, VEGFR-1, EGFR, VEGFR-2, IGF-1R, PDGFRα, and CD40 were identified in descending order as the most suitable biomarkers for NIRF imaging according to the modified TASC-scoring system. However, as the category of STS comprises an extensive and heterogenous group of tumors, it was chosen to specify the most optimal target for three common subtypes with infiltrative growth that are characterized by high rates of local recurrence: MFS, USTS and SS. While TEM1 was the optimal target for MFS, both TEM1 and PDGFRα were concluded to be most promising for USTS. In SS EGFR was considered most promising, yet closely followed by TEM1, VEGFR-1, and PDGFRα. However, as the expression of biomarkers and its extent is often not certain, an evaluation of the expression of biomarkers in preoperative biopsies could assist in designating the appropriate tracer for every patient. More importantly, for their potential use in NIRF imaging, data on contrast of expression on malignant and adjacent normal tissue is needed. Altogether, this systematic review paves the way for implementing fluorescence-guided surgery to optimize STS treatment.

## Figures and Tables

**Figure 1 cancers-13-00557-f001:**
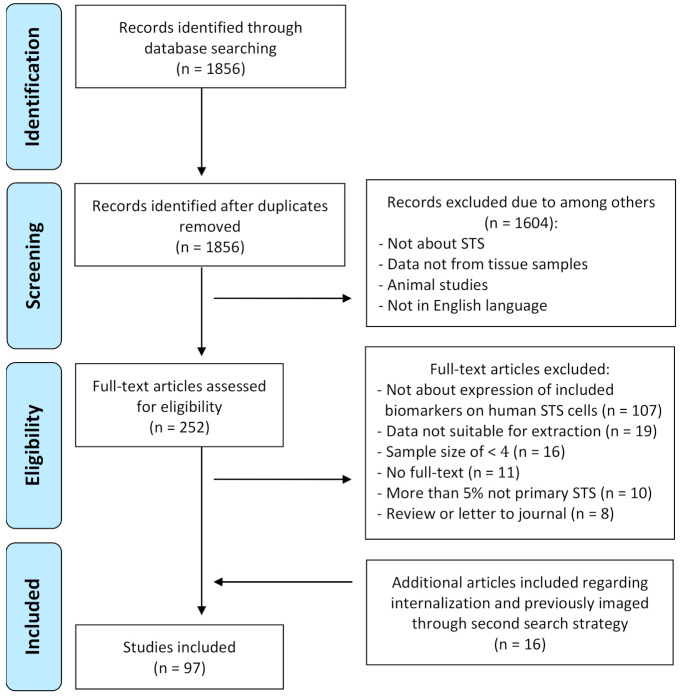
Flowchart of study selection process.

**Table 1 cancers-13-00557-t001:** Target selection scoring system.

Score	0	1	2
Sample size	0–100	101–500	>500
Percentage of positive STS samples	0–33%	33–67%	>67%
Pattern of expression *	Focal	Heterogeneous	Diffuse
Internalization	not described	Yes	
Previously imaged	not described	Yes, but not with NIRF imaging	Yes, NIRF imaging

Note. * Pattern of expression is focal when the expression is located in a specific region of the tumor sample and diffuse when expression is randomly spread throughout the tumor sample. When different samples show variable expression patterns (diffuse and focal), the expression pattern is defined as heterogeneous.

**Table 2 cancers-13-00557-t002:** Summarized data regarding eleven reviewed biomarkers (in descending order of the modified target selection criteria score).

Biomarker	Therapeutic Antibody	N	% Positive STS (Mean% + Range)	Pattern of Expression	Internalization	Previously Imaged	Score	Literature
Tumor endothelial marker 1 (TEM1/ endosialin/ CD248)	Ontuxizumab (MORAb-004)	768	77% (55–100)	Diffuse	Yes, [25]	NIRF imaging [26]	9	[26,27,28,29]
Vascular endothelial growth factor receptor-1 (VEGFR-1)	Aflibercept Bevacizumab	477	76% (22–100)	Diffuse	Yes, [30]	NIRF imaging [30,31]	8	[32,33,34,35,36,37,38,39]
Epidermal growth factor receptor (EGFR)	Cetuximab	1918	53% (0–100)	Diffuse	Yes, [40]	NIRF imaging [41]	8	[27,42,43,44,45,46,47,48,49,50,51,52,53,54,55,56,57,58,59,60,61,62,63,64,65,66,67,68,69,70,71,72,73,74,75,76]
Vascular endothelial growth factor receptor-2 (VEGFR-2)	Aflibercept Bevacizumab Ramucirumab	449	71% (11–100)	Diffuse	Yes, [77]	NIRF imaging [78]	7	[33,34,35,36,38,39,79,80,81]
Insulin-like growth factor 1 receptor (IGF-1R)	Ganitumab (AMG 479) Teprotumumab Cixutumumab Figitumumab	507	63% (25–100)	Diffuse	Yes, [82]	NIRF imaging [83]	7	[63,64,82,84,85,86,87,88,89]
Platelet derived growth factor receptor α (PDGFRα)	Olaratumab	1536	64% (0–100)	Diffuse	Yes, [84]	NIRF imaging [85]	7	[27,34,36,38,42,43,44,45,46,47,48,49,51,82,86,87,88,89,90,91,92]
Cluster of differentiation 40 (CD40)	APX005M	153	62% (17–86)	Diffuse	Yes, [93]	NIRF imaging [94]	7	[95,96,97,98]
Programmed death-ligand 1 (PD-L1/CD 274/B7-H1)	Atezolizumab Avelumab Durvalumab Envafolimab	1492	31% (0–76)	Heterogeneous (focal and diffuse)	Yes, [99]	NIRF imaging [100]	6	[101,102,103,104,105,106,107,108,109,110,111,112,113,114,115,116,117,118]
Leucine-rich repeat containing 15 (LRRC15)	ABBV-085	635	40%	Diffuse	Not described	Not described	4	[102]
Receptor tyrosine kinase-like orphan receptor 2 (ROR2)	CAB-ROR2-ADC	237	72%	Not described	Not described	Not described	3	[119]
Cytotoxic T-Lymphocyte-associated protein 4 (CTLA-4/CD152)	Ipilimumab Tremelimumab	10	30%	Not described	Yes, [120]	Not with NIRF imaging [120]	2	[59]

Note. Abbreviations: N, total number of samples; STS, soft tissue sarcoma; NIRF, near-infrared fluorescence

**Table 3 cancers-13-00557-t003:** Summarized data regarding biomarkers in myxofibrosarcoma, undifferentiated soft tissue sarcoma, and synovial sarcoma.

Biomarker	N	Positive Tumors Mean% (Range)	Expression Pattern	Present after RTx	Literature
Myxofibrosarcoma					
TEM1	34	100 (100)	Diffuse	Yes, [27]	[27]
EGFR	97	38 (0–89)	Heterogeneous	Yes, [27]	[26,53,65]
PDGFRα	34	77 (77)	Not described	Yes, [27]	[27]
Undifferentiated soft tissue sarcoma					
TEM1	128	81 (73–89)	Diffuse	N.D.	[28,29]
VEGFR-1	81	68 (68)	Not described	N.D.	[36]
EGFR	287	62 (5–95)	Heterogeneous	N.D.	[50,57,65,70]
VEGFR-2	81	6 (6)	Not described	N.D.	[36]
IGF-1R	120	25 (25)	Not described	N.D.	[90]
PDGFRα	432	79 (63–99)	Diffuse	N.D.	[35,50,126]
Synovial sarcoma					
TEM1	70	71 (62–80)	Heterogeneous	N.D.	[28,29]
VEGFR-1	27	70 (70)	Not described	N.D.	[27]
EGFR	160	86 (71–100)	Heterogeneous	Yes, [127]	[52,58,66,69,70,71]
VEGFR-2	27	4 (4)	Not described	N.D.	[27]
IGF-1R	195	57 (35–80)	Not described	N.D.	[81,82,128,129]
PDGFRα	136	69 (44–84)	Not described	N.D.	[35,81,88,91]

Abbreviations: N, total number of samples and/or cell lines; STS, soft tissue sarcoma; RTx, radiotherapy; N.D. not described. No distinction was made between monophasic and biphasic synovial sarcoma.

## Data Availability

No new data were created or analyzed in this study. Data sharing is not applicable to this article.

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
