# Peer review of "Candidate Biomarkers for Specific Intraoperative Near-Infrared Imaging of Soft Tissue Sarcomas: A Systematic Review"

_cancers, 2021, doi:10.3390/cancers13030557_

Round 1

Reviewer 1 Report

  • The paper is too long, which makes it difficult to read and makes the reader lose focus. Please see my specific comments below on how it might me shortened.
  • The introduction should be more concise. The problem of incomplete resection / contaminated resection margins with its associated poor prognosis can be summarised in two to three sentences without losing relevant information. Moreover, the last paragraph of the introduction, where you describe your search approach, should be moved to the methods. 
  • The term internalization should be defined when it's used the first time in the text. 
  • I suggest inclduing the expression of the respective biomarker in healthy tissue and thus the tumor-to-background ratio, into the TASC. As you rightfully point out in the discussion, this aspect is crucial when assessing the utility of a biomarker for imaging use. 
  • The description of the single biomarkers' properties and the corresponding clinical results when they are used for treatment should be shortened. A lot of this information is not directly relevan for the research question, and the way it is now, the manuscript is somewhat diffcult and tiring to read. 
  • Drawing safety information for the single biomarkers from clinical trials, in which the respective drug is given repeatedly or combined with chemotherapy, yields limited and somewhat biased infomation when this should be transfered to a situation where the compound is given once. Moreover, as you write, the doses used for intraoperative imaging are not comparable to the doses used for treatment. I suggest you look for safety information from dose-finding studies or first in human studies, where the compound is given only once. This aspect should also be addressed in the discussion.
  • The parts of the discusison and conclusions where appropriate biomarkers for the single STS subtypes are addressed is too long. In general, dividing STS by "conventional" subtypes for the purpose of antibodies and  fluorescence imaging is debatable, as these subtypes are usually not defined according to target expression. Since they are widely used in clinical practice, the subtypes can probably not be completely disregarded, but the pertinent discussion should take less space. 

Reviewer 2 Report

In this paper, the authors provide a systematic review of candidate near-infrared biomarkers  for soft tissue sarcomas. The papers includes both a search of literature and clinical trials. The paper is clearly written, well-structured, and interesting. I have some specific minor concerns that are listed below:

  • I would suggest to split the first paragraph of the “Introduction”, eg. I would recommend to start a new paragraph from “Three important…” (line 84).
  • A paragraph on molecular biology of soft tissue sarcomas in the “Introduction”  would helpful.
  • The authors provided scoring according to Target Selection Criteria. However, I would recommend to include also evaluation of biask risk, eg. according to Cochrane handbook http://handbook-5-1.cochrane.org/ . Some examples of such bias risk assessment can be found at https://journals.plos.org/plosone/article/figures?id=10.1371/journal.pone.0121388
  • Line 157 - ”The scoring system is based on the Target Selection Criteria (TASC) and its modified version developed by Bosma et al. [24,25].” – only ref. 24 is Bosma et al. – please correct
  • Search criteria for selection of the targets in the clinical trials databases should also be provided.
  • Line 190 “Our analysis of the EU Clinical Trials Register (https://www.clinicaltrialsregister.eu/) and clinical trials.gov (https://clinicaltrials.gov/) revealed the following clinically available monoclonal antibodies targeting STS-associated cell surface-expressed biomarkers (Table 1):” – please check if this a reference to the right table
  • Line 265 – “Besides, Thway et al.…” – missing citation number
  • Line 417 – please add citations in this paragraph
  • Line 449 – please place the reference numbers directly after the names of the authors
  • Line 520 “demonstrated” – please use another word
  • Line 548 – consider removing scores after the targets or explain that you are presenting them after each target
